# Self-Supervised Generative Adversarial Compression

**Chong Yu**
NVIDIA
chongy@nvidia.com

**Jeff Pool**
NVIDIA
jpool@nvidia.com

## Abstract

Deep learning's success has led to larger and larger models to handle more and more complex tasks; trained models often contain millions of parameters. These large models are compute- and memory-intensive, which makes it a challenge to deploy them with latency, throughput, and storage constraints. Some model compression methods have been successfully applied to image classification and detection or language models, but there has been very little work compressing generative adversarial networks (GANs) performing complex tasks. In this paper, we show that a standard model compression technique, weight pruning and knowledge distillation, cannot be applied to GANs using existing methods. We then develop a self-supervised compression technique which uses the trained discriminator to supervise the training of a compressed generator. We show that this framework has compelling performance to high degrees of sparsity, can be easily applied to new tasks and models, and enables meaningful comparisons between different compression granularities.

## 1 Introduction

Deep Neural Networks (DNNs) have been successful in various tasks like computer vision, natural language processing, recommendation systems, and autonomous driving. Modern networks are comprised of millions of parameters, requiring significant storage and computational effort. Though accelerators such as GPUs make realtime performance more accessible, compressing networks for faster inference and simpler deployment is an active area of research. Compression techniques have been applied to many networks to reduce memory requirements and improve performance. Though these approaches do not always harm accuracy, aggressive compression can adversely affect the behavior of the network. Distillation [1, 2] can improve the accuracy of a compressed network by using information from the original, uncompressed network.

Generative Adversarial Networks (GANs) [3, 4] are a class of DNN that consist of two sub-networks: a generative model and a discriminative model. Their training process aims to achieve a Nash Equilibrium between these two sub-models. GANs have been used in semi-supervised and unsupervised learning areas, such as fake dataset synthesis [5, 6], style transfer [7, 8], and image-to-image translation [9, 10]. Like networks used in other tasks, GANs have millions of parameters and nontrivial computational requirements.

In this work, we explore compressing the generative model of GANs for efficient deployment. We show that applying standard pruning techniques causes the generator's behavior to no longer achieve the network's goal and that past work targeted at compressing GANs for simple image synthesis fall short when they are applied to pruning large tasks. In some cases, this result is masked by loss curves that look identical to the original training. By modifying the loss function with a novel combination of the pre-trained discriminator and the original and compressed generators, we overcome this behavioral degradation and achieve compelling compression rates with little change in the quality of the compressed generator's ouput. We apply our technique to several networks and tasks to show generality. Finally, we study the behavior of compressed generators when pruned with different

amounts and types of sparsity, finding that a technique commonly used for accelerating image classification networks is not trivially applicable to GANs, but a recently-introduced fine-grained structured sparsity is quite successful.

Our main contributions are:

- We illustrate that and explain why compressing the generator of a GAN with existing methods is unsatisfactory for complex tasks. (Section 3)
- We propose self-supervised compression for the generator in a GAN. (Section 4)
- We show that our technique can apply to several networks and tasks. (Section 5)
- We show and analyze qualitative differences in compression ratio and granularity. (Section 6)

## 2  Related research

A common method of DNN compression is network pruning [11]: setting the small weights of a trained network to zero and fine-tuning the remaining weights to recover accuracy. Zhu & Gupta [12] proposed a gradual pruning technique (AGP) to compress the model during the initial training process. Wen et al. [13] proposed a structured sparsity learning method that uses group regularization to force weights towards zero, leading to pruning groups of weights together. Li et al. [14] pruned entire filters and their connecting feature maps from models, allowing the network to run with standard dense software libraries. Though it was initially applied to image classification networks, network pruning has been extended to natural language processing tasks [15, 16] and to recurrent neural networks (RNNs) of all types - vanilla RNNs, GRUs [17], and LSTMs [18]. As with classification networks, structured sparsity within recurrent units has been exploited [19].

A complementary method of network compression is quantization. Sharing weight values among a collection of similar weights by hashing [20] or clustering [21] can save storage and bandwidth at runtime. Changing fundamental data types adds the ability to accelerate the arithmetic operations, both in training [22] and inference regimes [23].

Several techniques have been devised to combat lost accuracy due to compression, since there is always the chance that the behavior of the network may change in undesirable ways when the network is compressed. Using GANs to generate unique training data [24] and extracting knowledge from an uncompressed network, known as distillation [2], can help keep accuracy high. Since the pruning process involves many hyperparameters, Lin et al. [25] use a GAN to guide pruning, and Wang et al. [26] structure compression as a reinforcement learning problem; both remove some user burden.

## 3  Existing techniques fail for a complex task

Though there are two networks in a single GAN, the main workload at deployment is usually from the generator. For example, in image synthesis and style transfer tasks, the final output images are created solely by the generator. The discriminator is vital in training, but it is abandoned afterward for many tasks. So, when applying state-of-the-art compression methods to GANs, we focus on the generator for efficient deployment. We look at two broad categories of baseline approaches: standard pruning techniques that have been applied to other network architectures, and techniques that were devised to compress the generator of a GAN performing image synthesis. We compare the dense baseline [a] to our technique [b], as well as a small, dense network with the same number of parameters [c]. (Labels correspond to entries in Table 1, the overview of all techniques, and Figure 1, results of each technique).

**Standard Pruning Techniques**. To motivate GAN-specific compression methods, we try variations of two state-of-the-art pruning methods: manually pruning and fine tuning [11] a trained dense model [d], and AGP [12] from scratch [e] and during fine-tuning [f]. We also include distillation [2] to improve the performance of the pruned network with manual pruning [g] and AGP fine-tuning [h]. Distillation is typically optional for other network types, since it is possible to get decent accuracy with moderate pruning in isolation. For very aggressive compression or challenging tasks, distillation aims to extract knowledge for the compressed (student) network from original (teacher) network's behavior. We also fix the discriminator of [g] to see if the discriminator was being weakened by the compressed generator [i].

Table 1: GAN compression algorithms comparison

| Technique | Generator(s) Compressed | Generator(s) Init Scheme | Discriminator Init Scheme | Fixed | L-Gc | L-Dc | L-Go | L-Do | Results Qualitative | FID Score |
|---|---|---|---|---|---|---|---|---|---|---|
| (a) No Compression [10] | Dense | Random | Dense,Random | No | - | - | Yes | Yes | Good | 6.113 |
| (b) Self-Supervised MIS **(ours)** | Dense,Sparse | From Dense | Dense,Pretrained | No | Yes | Yes | Yes | Yes | Good | 6.929 |
| (c) Small & Dense Network | Dense | Random | Dense,Random | No | - | - | Yes | Yes | Mode collapse | 72.821 |
| (d) One-shot Pruning & Fine-Tuning [11] | Sparse | From Dense | Dense,Pretrained | No | Yes | Yes | - | - | Facial artifacts | 24.404 |
| (e) Gradual Pruning & Fine-Tuning [12] | Sparse | From Dense | Dense,Random | No | Yes | Yes | - | - | Facial artifacts | 35.677 |
| (f) Gradual Pruning during Training [12] | Sparse | Random | Dense,Random | No | Yes | Yes | - | - | No faces | 84.941 |
| (g) One-shot Pruning & Distillation [2] | Dense,Sparse | From Dense | - | - | Yes | - | Yes | - | Mode collapse | 45.461 |
| (h) (d) & Distillation [2] | Dense,Sparse | From Dense | Dense,Pretrained | No | Yes | Yes | Yes | - | Color artifacts | 38.985 |
| (i) (g) & Fix Original Loss | Dense,Sparse | From Dense | Dense,Pretrained | Yes | Yes | Yes | - | - | Facial artifacts | 15.182 |
| (j) Adversarial Learning [27] | Dense,Sparse | Random | Dense,Random | No | Yes | Yes | Yes | Yes | Mode collapse | 92.721 |
| (k) Knowledge Distillation [28] | Dense,Sparse | From Dense | Dense,Random | No | Yes | - | Yes | Yes | Mode collapse | 103.094 |
| (l) Distill Intermediate (LIT) [29] | Dense,Sparse | From Dense | Dense,Pretrained | Yes | - | - | - | - | Mode collapse | 61.150 |
| (m) E-M Pruning [30] | Dense,Sparse | From Dense | Sparse,Pretrained | No | - | Yes | Yes | - | Color artifacts | 159.767 |
| (n) G & D Both Pruning [31] | Dense,Sparse | From Dense | Sparse,Pretrained | No | Yes | Yes | Yes | - | Mode collapse | 46.453 |

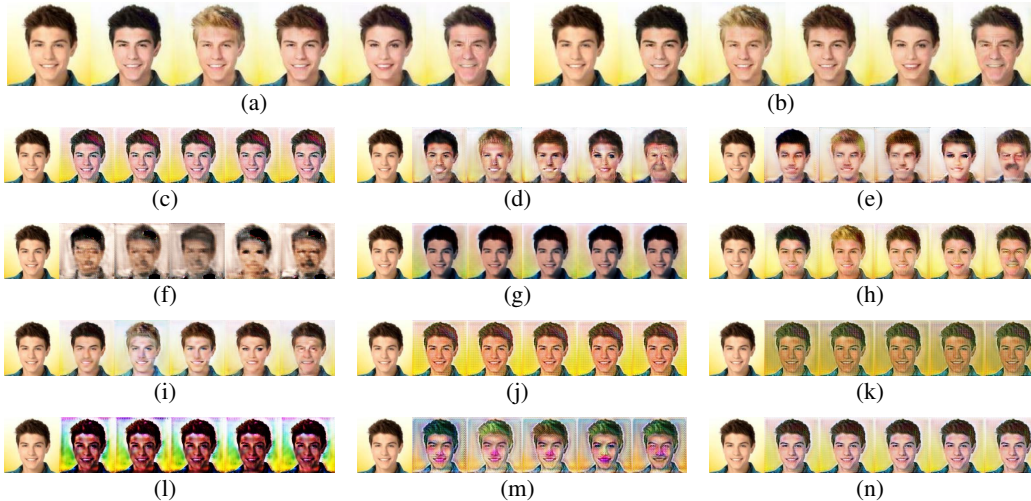

Figure 1: Various approaches to compress StarGAN (CelebA data set) with network pruning. Each group shows one input face translated with different methods of compressing the network: **a**. Uncompressed, **b**. Self-Supervised **(ours)**, **c**. Small and dense, **d**. One-shot pruning and fine-tuning, **e**. AGP as fine-tuning, **f**. AGP from scratch, **g**. One-shot pruning and distilling, **h**. AGP during distillation, **i**. AGP during distillation with fixed discriminator, **j**. Adversarial learning, **k**. Knowledge distillation, **l**. Distillation on output of intermediate layers, **m**. E-M pruning, and **n**. Prune both G and D models.

**Targeted GAN Compression**. There has been some work in compressing GANs with methods other than pruning. For this category, we decompose each instance of prior work into two areas: the method of compression (e.g. quantization, layer removal, etc.) and the modifications required to make the compression succeed (e.g. distillation, novel training schemes, etc.). For comparisons to these techniques, we apply the modifications presented in prior research to the particular method of compression on which we focus, network pruning. We first examine two approaches similar to ours. Adversarial training [27] [j] posits that during distillation of a classification network, the student network can be thought of as a generative model attempting to produce features similar to that of the teacher model. So, a discriminator was trained alongside the student network, trying to distinguish between the student and the teacher. One could apply this technique to compress the generator of a GAN, but we find that its key shortcoming is that it trains a discriminator from scratch. Similarly, distillation has been used to compress GANs [28] [k], but again, the "teacher" discriminator was not used when teaching the "student" generator.

Learned Intermediate Representation Training (LIT) [29] [l] compresses StarGAN by a factor of $1.8\times$ by training a shallower network. Crucially, LIT does not use the pre-trained discriminator in any loss function. Quantized GANs (QGAN) [30] [m] use a training process based on Expectation-Maximization to achieve impressive compression results on small generative tasks with output images of 32x32 or 64x64 pixels. Liu et al. [31] find that maintaining a balance between discriminator and generator is key: their approach is to selectively binarize parts of both networks in the training process on the CelebA generative task. So, we try pruning both networks during the training process [n].

**Experiments**. For these experiments, we use StarGAN[1] [10] trained with the *Distiller* [32] library for the pruning. StarGAN extends the image-to-image translation capability from two domains to multiple domains within a single unified model. It uses the CelebFaces Attributes (CelebA) [33] as the dataset. CelebA contains 202,599 images of celebrities' faces, each annotated with 40 binary attributes. As in the original work, we crop the initial images from size $178 \times 218$ to $178 \times 178$, then resize them to $128 \times 128$ and randomly select 2,000 images as the test dataset and use remaining images for training. The aim of StarGAN is facial attribute translation: given some image of a face, it generates new images with five domain attributes changed: 3 different hair colors (black, blond, brown), different gender (male/female), and different age (young/old). Our target sparsity is 50% for each approach.

We stress that we attempted to find good hyperparameters when using the existing techniques, but standard approaches like reducing the learning rate for fine-tuning [11], etc., were not helpful. Further, the target sparsity, 50%, is not overly aggressive, other tasks readily achieve 80%-90% fine-grained sparsity with minimal accuracy impact.

The results of these trials are shown in Figure 1. Subjectively, it is easy to see that the existing approaches (1(c) through 1(n)) produce inferior results to the original, dense generator. Translated facial images from pruning & naïve fine-tuning (1(d) and 1(e)) do give unique results for each latent variable, but the images are hardly recognizable as faces. These fine-tuning procedures, along with AGP from scratch (1(f)) and distillation from intermediate representations (1(l)), simply did not converge. One-shot pruning and traditional distillation (1(g)), adversarial learning (1(j)), knowledge distillation (1(k)), training a "smaller, dense" half-sized network from scratch (1(c)) and pruning both generator and discriminator (1(n)) keep facial features intact, but the image-to-image translation effects are lost to mode collapse (see below). There are obvious mosaic textures and color distortion on the translated images from fine-tuning & distillation (1(h)), without fine-tuning the original loss (1(i)), and from the pruned model based on the Expectation-Maximization (E-M) algorithm (1(m)). On the other hand, the translated facial images from a generator compressed with our proposed self-supervised GAN compression method (1(b)) are more natural, nearly indistinguishable from the dense baseline (1(a)), and match the quantitative Frechet Inception Distance (FID) scores [34] in Table 1. While past approaches have worked to prune some networks on other tasks (DCGAN generating MNIST digits, see A.2 in the *Appendix*), we show that they do not succeed on larger image-to-image translation tasks, while our approach works on both. Similarly, though LIT [29] [l] was able to achieve a compression rate of $1.8\times$ on this task by training a shallower network, it does not see the same success at network pruning with a higher rate.

**Discussion**. It is tempting to think that the loss curves of the experiment for each technique can tell us if the result is good or not. We found that for many of these experiments, the loss curves correctly predicted that the final result would be poor. However, the curves for [h] and [m] look very good - the compressed generator and discriminator losses converge at 0, just as they did for baseline training. It is clear from the results of querying the generative models (Figures 1(h) and 1(m)), though, that this promising convergence is a false positive. In contrast, the curves for our technique predict good performance, and, as we prune more aggressively in Section 6, higher loss values correlate well with worsening FID scores. (Loss curves are provided in A.1 and A.8 in the *Appendix*.)

As pruning and distillation are very effective when compressing models for image classification tasks, why do they fail to compress this generative model? We share three potential reasons:

1. Standard pruning techniques need explicit evaluation metrics; softmax easily reflects the probability distribution and classification accuracy. GANs are typically evaluated subjectively, though some imperfect quantitative metrics have been devised.
2. GAN training is relatively unstable [35, 31] and sensitive to hyperparameters. The generator and discriminator must be well-matched, and pruning can disrupt this fine balance.
3. The energy of the input and output of a GAN is roughly constant, but other tasks, such as classification, produce an output (1-hot label vector) with much less entropy than the input (three-channel color image of thousands of pixels).

Elaborating on this last point, there is more tolerance in the reduced-information space for the compressed classification model to give the proper output. That is, even if the probability distribution inferred by the original and compressed classification models are not exactly the same, the classified labels *can* be the same. On the other hand, tasks like style-transfer and dataset synthesis have no

obvious energy reduction. We need to keep entropy as high as possible [36] during the compression process to avoid mode collapse – generating the same output for different inputs or tasks. Attempting to train a new discriminator to make the compressed generator behave more like the original generator [27] suffers from this issue – the new discriminator quickly falls into a low-entropy solution and cannot escape. Not only does this preclude its use on generative tasks, but it means that the compressed network for any task must also be trained from scratch during the distillation process, or the discriminator will never be able to learn.

## 4 Self-Supervised generator compression

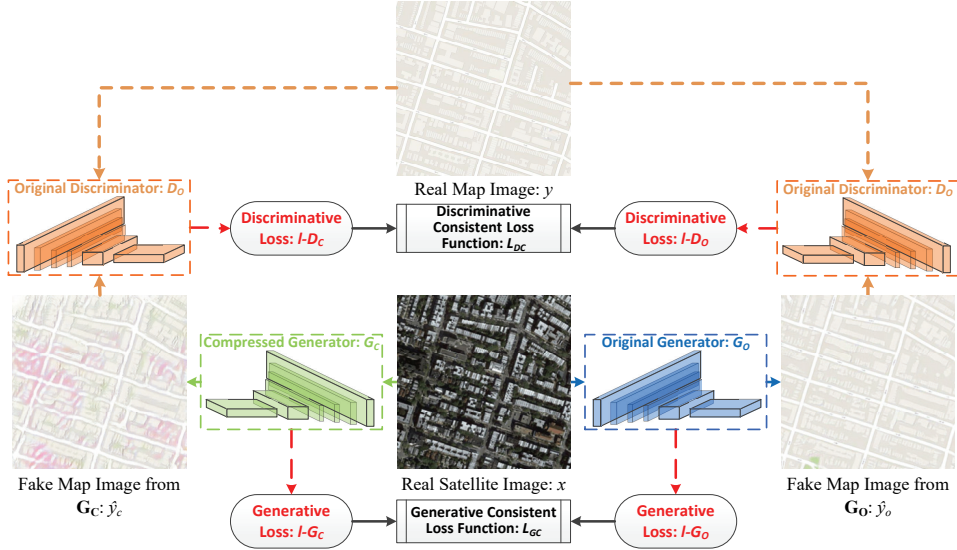

Figure 2: Workflow of self-supervised (by the original discriminator) GAN compression.

We seek to solve each of the problems highlighted above. Let us restate the general formulation of GAN training: the purpose of the generative model is to generate new samples which are very similar to the real samples, but the purpose of the *discriminative* model is to distinguish between real samples and those synthesized by the generator. A fully-trained discriminator is good at spotting differences, but a well-trained generator will cause it to believe that the a generated sample is both real and generated with a probability of 0.5. Our main insight follows:

By using this powerful discriminator that is already well-trained on the target data set, we can allow it to stand in as a quantitative subjective judge (point 1, above) – if the discriminator can't tell the difference between real data samples and those produced by the compressed generator, then the compressed generator is of the same quality as the uncompressed generator. A human no longer needs to inspect the results to judge the quality of the compressed generator. This also addresses our second point: by starting with a trained discriminator, we know it is well-matched to the generator and will not be overpowered. Since it is so capable (there is no need to prune it too), it also helps to avoid mode collapse. As distillation progresses, it can adapt to and induce fine changes in the compressed generator, which is initialized from the uncompressed generator. Since the original discriminator is used as a proxy for a human's subjective evaluation, we refer to this as "self-supervised" compression.

We illustrate the workflow in Figure 2, using a GAN charged with generating a map image from a satellite image in a domain translation task. In the right part of Figure 2, the real satellite image ($x$) goes through the original generative model ($G_O$) to produce a fake map image ($\hat{y}_o$). The corresponding generative loss value is $l$-$G_O$. Accordingly, in the left part of Figure 2, the real satellite image ($x$) goes through the compressed generative model ($G_C$) to produce a fake map image ($\hat{y}_c$). The corresponding generative loss value is $l$-$G_C$. The inference process of the original and compressed generators are expressed as follows:

$$\hat{y}_o = G_O(x), \qquad \hat{y}_c = G_C(x) \qquad (1)$$

The overall generative difference is measured between the two corresponding generative losses. We use a generative consistent loss function ($L_{GC}$) in the bottom of Figure 2 to represent this process.

$$L_{GC}(l\text{-}G_O, l\text{-}G_C) \to 0 \qquad (2)$$

Since the GAN training process aims to reduce the differences between real and generated samples, we stick to this principle in the compression process. In the upper right of Figure 2, real map image ($y$) and fake map image ($\hat{y}_o$) go through the original discriminative model $\boldsymbol{D}_O$. $\boldsymbol{D}_O$ tries to ensure that the distribution of $\hat{y}_o$ is indistinguishable from $y$ using an adversarial loss. The corresponding discriminative loss value is $l\text{-}\boldsymbol{D}_O$. In the upper left of Figure 2, real map image ($y$) and fake map image ($\hat{y}_c$) also go through the original discriminative model $\boldsymbol{D}_O$. In this way, we use the original discriminative model as a "self-supervisor." The corresponding discriminative loss value is $l\text{-}\boldsymbol{D}_C$.

$$l\text{-}\boldsymbol{D}_O = \boldsymbol{D}_O(y, \hat{y}_o), \qquad l\text{-}\boldsymbol{D}_C = \boldsymbol{D}_O(y, \hat{y}_c) \tag{3}$$

So the discriminative difference is measured between two corresponding discriminative losses. We use the discriminative consistent loss function $\boldsymbol{L}_{DC}$ in the top of Figure 2 to represent this process.

$$\boldsymbol{L}_{DC}(l\text{-}\boldsymbol{D}_O, l\text{-}\boldsymbol{D}_C) \to 0 \tag{4}$$

The generative and discriminative consistent loss functions ($\boldsymbol{L}_{GC}$ and $\boldsymbol{L}_{DC}$) use the weighted normalized distance. Taking the **StarGAN** task as the example (other tasks may use different losses)[2]:

$$\boldsymbol{L}_{GC}(l\text{-}\boldsymbol{G}_O, l\text{-}\boldsymbol{G}_C) = \frac{|l\text{-}\boldsymbol{Gen}_O - l\text{-}\boldsymbol{Gen}_C|}{|l\text{-}\boldsymbol{Gen}_O|} + \alpha \frac{|l\text{-}\boldsymbol{Cla}_O - l\text{-}\boldsymbol{Cla}_C|}{|l\text{-}\boldsymbol{Cla}_O|} + \beta \frac{|l\text{-}\boldsymbol{Rec}_O - l\text{-}\boldsymbol{Rec}_C|}{|l\text{-}\boldsymbol{Rec}_O|} \tag{5}$$

where $l\text{-}\boldsymbol{Gen}$, $l\text{-}\boldsymbol{Cla}$ and $l\text{-}\boldsymbol{Rec}$ is the generation, classification and reconstruction loss term, respectively. $\alpha$ and $\beta$ are the weight ratios among loss terms. (We use the same values as StarGAN baseline.)

$$\boldsymbol{L}_{DC}(l\text{-}\boldsymbol{D}_O, l\text{-}\boldsymbol{D}_C) = |l\text{-}\boldsymbol{Dis}_O - l\text{-}\boldsymbol{Dis}_C|/|l\text{-}\boldsymbol{Dis}_O| + \delta |l\text{-}\boldsymbol{GP}_O - l\text{-}\boldsymbol{GP}_C|/|l\text{-}\boldsymbol{GP}_O| \tag{6}$$

where $l\text{-}\boldsymbol{Dis}$ is the discriminative loss item, $l\text{-}\boldsymbol{GP}$ is the gradient penalty loss item, and $\delta$ is a weighting factor (again, we use the same value as the baseline).

The overall loss function of GAN compression consists of generative and discriminative losses.

$$L_{Overall} = \boldsymbol{L}_{GC}(l\text{-}\boldsymbol{G}_O, l\text{-}\boldsymbol{G}_C) + \lambda \boldsymbol{L}_{DC}(l\text{-}\boldsymbol{D}_O, l\text{-}\boldsymbol{D}_C), \tag{7}$$

where $\lambda$ is the parameter to adjust the percentages between generative and discriminative losses.

We showed promising results with this method above in the context of prior methods. In the following experiments, we investigate how well the method applies to other networks and tasks (Section 5) and how well the method works on different sparsity ratios and pruning granularities (Section 6) .

## 5 Application to new tasks and networks

For experiments in this section, we prune individual weights in the generator. The final sparsity rate is 50% for all convolution and deconvolution layers in the generator (unless otherwise noted, and more aggressive sparsities are discussed in Section 6). Following AGP [12], we gradually increase the sparsity from 5% at the beginning to our target of 50% halfway through the self-supervised training process, and we set the loss adjustment parameter $\lambda$ to 0.5 in all experiments. We use *PyTorch* [37], implement the pruning and training schedules with *Distiller* [32], and train and generate results with V100 GPU [38] to match public baselines. In all experiments, the data sets, data preparation, and baseline training all follow from the public repositories - details are summarized in Table 2. We start by assuming an extra 10% of the original number of epochs will be required; in some cases, we reduced the overhead to only 1% while maintaining subjective quality. We include representative results for each task; a more comprehensive collection of outputs is included in the ***Appendix***.

Table 2: Tasks and networks overview

| Task | Network | Dataset | Resolution | FID Scores when Pruned to | | | | |
|---|---|---|---|---|---|---|---|---|
| | | | | 0% (dense) | 25% | 50% | 75% | 90% |
| Image Synthesis | DCGAN | MNIST | 64x64 | 50.391 | 50.128 | 50.634 | 50.805 | 51.356 |
| Domain Translation | Pix2Pix | Sat $\to$ Map | 256x256 | 17.636 | 17.897 | 17.990 | 20.235 | 24.892 |
| Domain Translation | Pix2Pix | Sat $\leftarrow$ Map | 256x256 | 30.826 | 30.628 | 30.720 | 34.051 | 38.936 |
| Style Transfer | CycleGAN | Monet $\to$ Photo | 256x256 | 63.152 | 63.410 | 63.662 | 66.394 | 70.933 |
| Style Transfer | CycleGAN | Monet $\leftarrow$ Photo | 256x256 | 31.987 | 32.102 | 32.346 | 33.913 | 41.409 |
| Image-Image Translation | CycleGAN | Zebra $\to$ Horse | 256x256 | 60.930 | 61.005 | 61.102 | 65.898 | 68.450 |
| Image-Image Translation | CycleGAN | Zebra $\leftarrow$ Horse | 256x256 | 52.862 | 52.631 | 52.688 | 58.356 | 63.274 |
| Image-Image Translation | StarGAN | CelebA | 128x128 | 6.113 | 6.307 | 6.929 | 6.714 | 7.144 |
| Super Resolution | SRGAN | DIV2K | $\geq$ 512x512 | 14.653 | 15.236 | 16.609 | 17.548 | 18.376 |

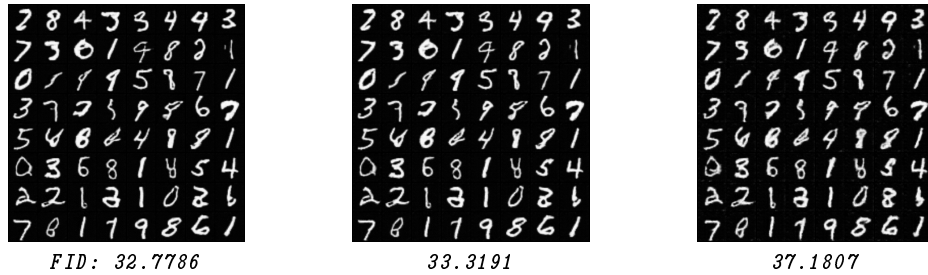

FID: 32.7786                    33.3191                    37.1807

Figure 3: Image synthesis on MNIST dataset with DCGAN. L-R: Handwritten numbers generated by the original generator and compressed generators with 50% and 75% fine-grained sparsity.

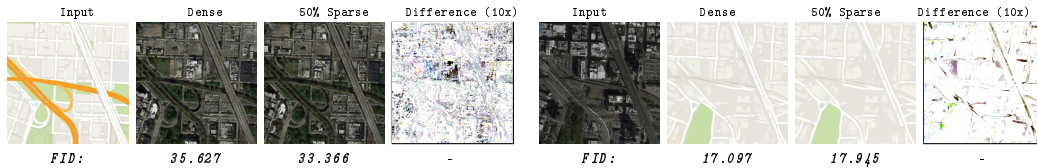

FID:        35.627        33.366        -                FID:        17.097        17.945        -

Figure 4: Representative results for domain translation: pix2pix. Columns 1-4: map to satellite task, Columns 5-8: satellite to map task.

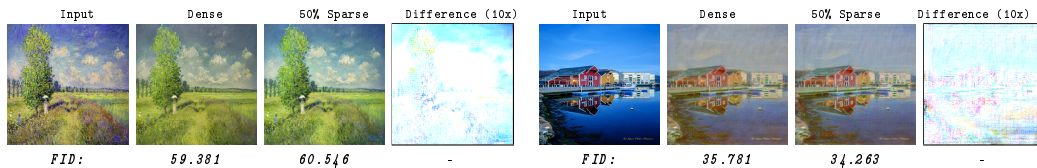

FID:        59.381        60.546        -                FID:        35.781        34.263        -

Figure 5: Representative results for style transfer: CycleGAN. Columns 1-4: Monet painting to photo, Columns 5-8: photo to Monet painting.

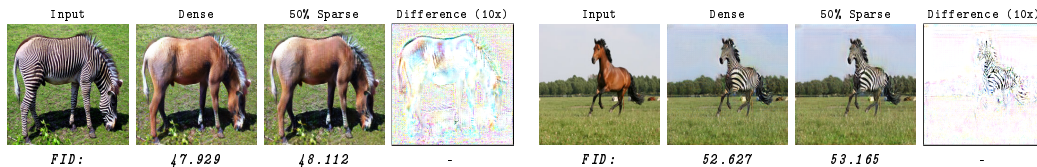

FID:        47.929        48.112        -                FID:        52.627        53.165        -

Figure 6: Representative image-to-image translation results: CycleGAN. Columns 1-4: zebra to horse, Columns 5-8: horse to zebra.

**Image Synthesis**. We apply the proposed compression method to DCGAN [5][3], a network that learns to synthesize novel images from some distribution. We task DCGAN with generating images that could belong to the MNIST data set, with results shown in Figure 3.

**Domain Translation**. We apply the proposed compression method to pix2pix [39][4], an approach to learn the mapping between paired training examples by applying conditional adversarial networks. In our experiment, the task is synthesizing fake satellite images from label maps and vice-versa. Representative results of this bidirectional task are shown in Figure 4.

**Style Transfer**. We apply the proposed compression method to CycleGAN [9], used to exchange the style of images from a source domain to a target domain in the absence of paired training examples. In our experiment, the task is to transfer the style of real photos with that of Monet's paintings. Representative results of this bidirectional task are shown in Figure 5: photographs are given the style of Monet's paintings and vice-versa.

**Image-to-image Translation**. In addition to the StarGAN results above (Section 3, Figure 1), we apply the proposed compression method to CycleGAN [9] performing bidirectional translation between images of zebras and horses. Results are shown in Figure 6.

Table 3: *PSNR* (dB), *SSIM* and *FID* indicators for validation datasets

| Dataset | Original Generator | | | 50% Filter-Compressed G | | | 50% Element-Compressed G | | | 90% Element-Compressed G | | |
|---|---|---|---|---|---|---|---|---|---|---|---|---|
| | PSNR | SSIM | FID | PSNR | SSIM | FID | PSNR | SSIM | FID | PSNR | SSIM | FID |
| Set5 | 31.063 | 0.853 | 30.762 | 30.234 | 0.860 | 39.514 | 30.484 | 0.862 | 36.824 | 30.301 | 0.861 | 37.475 |
| Set14 | 27.643 | 0.716 | 55.457 | 27.315 | 0.745 | 82.118 | 27.417 | 0.744 | 70.126 | 27.369 | 0.743 | 80.684 |
| DIV2K | 29.206 | 0.778 | 14.653 | 28.876 | 0.801 | 18.500 | 28.975 | 0.801 | 16.609 | 28.868 | 0.798 | 18.263 |

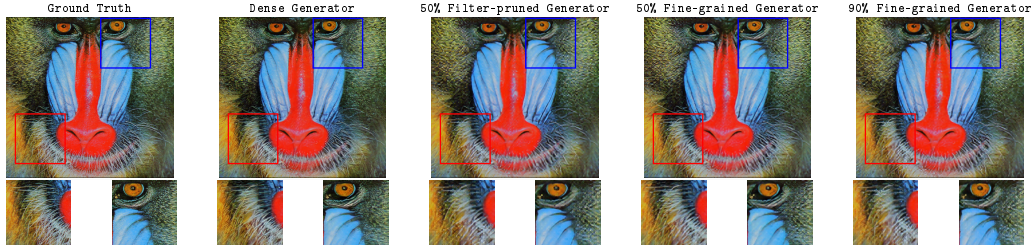

Figure 7: Representative super resolution results: SRGAN (with enlargements of boxed areas).

**Super Resolution**. We apply self-supervised compression to SRGAN [40][5], which uses a discriminator network trained to differentiate between upscaled and the original high-resolution images. We trained SRGAN on the DIV2K data set [41], and use the DIV2K validation images, as well as Set5 [42] and Set14 [43] to report deployment quality. In this task, quality is often evaluated by two metrics: Peak Signal-to-Noise Ratio (PSNR) [44] and Structural Similarity (SSIM) [45]. We also show FID scores [34] for our results in the results summarized in Table 3, and a representative output is shown in Figure 7. These results also include filter-pruned generators (see Section 6).

# 6 Effect of Compression Ratio and Granularity

After showing that self-supervised compression applies to many tasks and networks with a moderate, fine-grained sparsity of 50%, we explore ways to achieve a performance speedup: different pruning granularities and rates. Finer-grained sparsity results in higher accuracy, but pruning entire filters [14] results in a smaller, dense workload that is easy to accelerate. Similarly, higher sparsity can also increase runtime performance, but may affect network behavior.

We pruned all tasks by removing both single elements and entire filters. Further, for each granularity, we pruned to final sparsities of 25%, 50%, 75%, and 90%. Representative results for CycleGAN (Monet → Photo) and StarGAN are shown in Figure 8 and Figure 9, with results for all tasks in the *Appendix*. After up to 90% fine-grained sparsity, some fine details faded away in CycleGAN and StarGAN, but filter pruning results in drastic color shifts and loss of details at even 25% sparsity. Since filter pruning did not fare well, we also look at the recently-introduced 2:4 fine-grained structured sparsity, which can directly give a performance increase on the NVIDIA A100 GPU [46]. Results for this method (Table 4 and Figure 9) are indistinguishable from 50% unstructured sparsity, but simple to accelerate.

Table 4: Accuracy of pruning for a practical performance benefit.

| Task | Network | Dataset | FID Scores when Pruned to | | | | | |
|---|---|---|---|---|---|---|---|---|
| | | | (Dense) | (2:4) | (Filter Pruned) | | | |
| | | | 0% | 50% | 25% | 50% | 75% | 90% |
| Image Synthesis | DCGAN | MNIST | 50.391 | 50.535 | 60.154 | 81.014 | 103.134 | 135.066 |
| Domain Translation | Pix2Pix | Sat → Map | 17.636 | 17.892 | 21.655 | 29.324 | 39.863 | 63.475 |
| Domain Translation | Pix2Pix | Sat ← Map | 30.826 | 30.715 | 33.691 | 43.009 | 59.249 | 76.315 |
| Style Transfer | CycleGAN | Monet → Photo | 63.152 | 63.002 | 92.724 | 140.056 | 222.826 | 217.818 |
| Style Transfer | CycleGAN | Monet ← Photo | 31.987 | 32.357 | 81.517 | 105.812 | 204.113 | 182.336 |
| Image-Image Translation | CycleGAN | Zebra → Horse | 60.930 | 61.032 | 83.223 | 109.373 | 139.045 | 167.703 |
| Image-Image Translation | CycleGAN | Zebra ← Horse | 52.862 | 52.450 | 70.947 | 99.580 | 125.465 | 165.145 |
| Image-Image Translation | StarGAN | CelebA | 6.113 | 6.927 | 28.037 | 41.318 | 53.645 | 61.367 |
| Super Resolution | SRGAN | DIV2K | 14.653 | 16.605 | 16.928 | 18.499 | 20.923 | 23.093 |

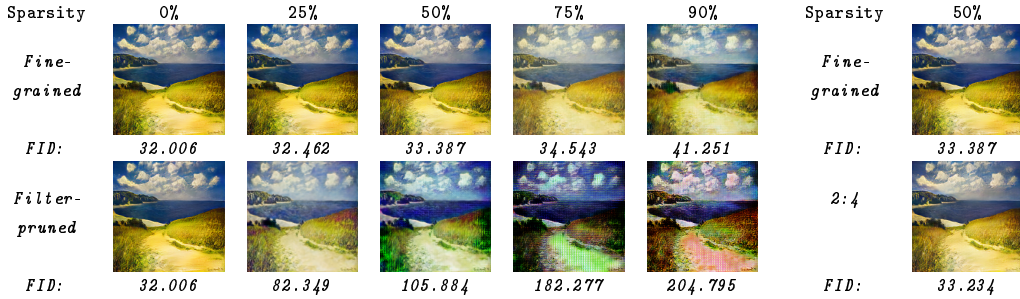

Figure 8: Representative results for pruning rate and structure study of style transfer.

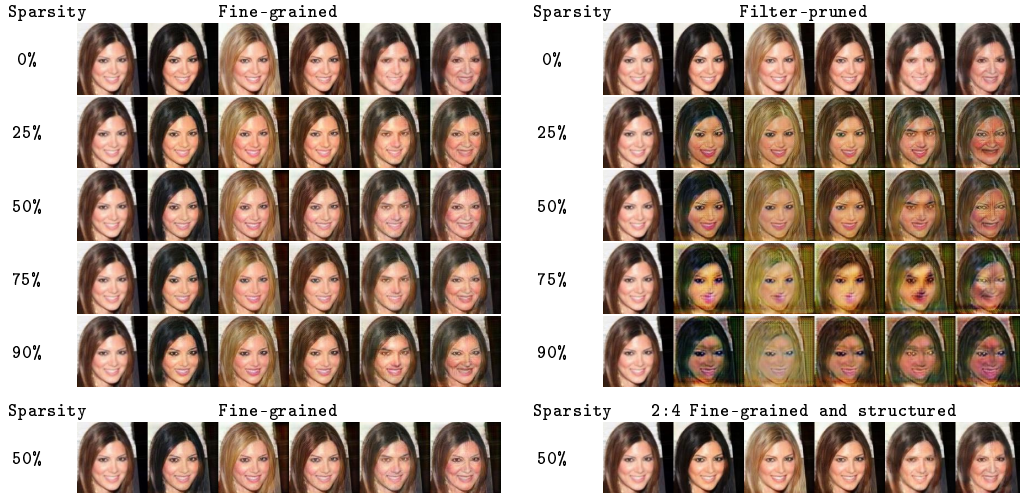

Figure 9: Representative results for pruning rate and structure of image-to-image translation.

# 7 Conclusion and Future Work

Network pruning has been applied to various networks, but never to GANs performing complex tasks. We showed that existing pruning approaches fail to retain network quality, as do training modifications aimed at compressing simple GANs by other methods applied to pruning. To solve this, we used a pre-trained discriminator to self-supervise the pruning of several GANs' generators and showed this method performs well both qualitatively and quantitatively. Advantages of our method include:

- The results from the compressed generators are greatly improved over past work.
- The self-supervised compression is much shorter than the original GAN training process - only 1-10% of the original training time is needed.
- It is an end-to-end compression schedule that does not require objective evaluation metrics; final quality is accurately reflected in loss curves.
- We introduce a single optional hyperparameter (fixed to 0.5 for all our experiments).

We use self-supervised GAN compression to show that pruning whole filters, which can work well for image classification models, may perform poorly for GAN applications. Even pruned at a moderate sparsity (e.g. 25% in Figure 8), the generated image has an obvious color shift and does not transfer the photorealistic style. In contrast, the fine-grained compression strategy works well for all tasks we explored, even when constrained to a structured 2:4 pattern.

Finally, we have not tried to achieve extremely aggressive compression rates with complicated pruning strategies. Different models may be able to tolerate different amounts of pruning when applied to a task, which we leave to future work. Similarly, while we have used network pruning to show the importance and utility of the proposed method, self-supervised compression is general to other techniques, such as quantization, weight sharing, etc. There are other tasks for which GANs can provide compelling results, and newer networks for tasks we have already explored; future work will extend our self-supervised method to these new areas.

## Broader Impact

In this paper, we propose a self-supervised compression technique for generative adversarial networks and prove its effectiveness across various typical and complex tasks. We also show the fine-grained compression strategy works better than coarse-grained compression methods.

Our proposed compression technique can benefit various applications for creative endeavors. Mobile applications performing style transfer or super-resolution on the client to save bandwidth can benefit from simpler generators. Artists may use inpainting or other texture-generation techniques to save asset storage space or interactive video generation to save rendering time, and musicians may want a backing track to generate novel accompaniment that responds in real-time.

GANs are also used to augment training data for tasks like autonomous driving, medical imaging, etc. Compressed models with higher deployment efficiency will help generate more valuable data to train more robust and accurate networks for pedestrian detection, emergency protection, medical analysis, and diagnosis. Further, a more efficient data augmentation solution will leave more resources available to train a more capable network. Our hope is that these effects eventually improve peoples' safety and well-being.

We also encourage researchers to understand and mitigate the risks arising from GAN applications. As a generative network has the power to change the style or content of paintings and photos, we should notice the risk that it can be used to misrepresent objective truth. However, we expect such misuse will become ineffectual as GAN and detection techniques improve; these techniques may similarly benefit from our contributions.

## Footnotes

[1]StarGAN baseline repository: `https://github.com/yunjey/StarGAN`.

[2]In different GANs, the generative loss may consist of several sub-items. For example, StarGAN combines adversarial loss, domain classification loss and reconstruction loss into overall generative loss.

[3]DCGAN baseline repository: `https://github.com/pytorch/examples/tree/master/dcgan`.

[4]Pix2pix, CycleGAN repository: `https://github.com/junyanz/pytorch-CycleGAN-and-pix2pix`.

[5]SRGAN baseline repository: `https://github.com/xinntao/BasicSR`.

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
