[Supplementary Material]

# Appendix: Self-Supervised Generative Adversarial Compression

**Chong Yu**
NVIDIA
chongy@nvidia.com

**Jeff Pool**
NVIDIA
jpool@nvidia.com

## A  Appendix

Since GANs are typically judged with subjective, qualitative observations, we present several results for each of the experiments in the main paper so readers can see the motivation for the conclusions drawn therein. We organize this document in the following way:

- Section A.1: Naïve Compression with StarGAN
- Section A.2: Image Synthesis with DCGAN
- Section A.3: Domain Translation with Pix2Pix
- Section A.4: Style Transfer with CycleGAN
- Section A.5: Image-Image Translation with CycleGAN
- Section A.6: Image-Image Translation with StarGAN
- Section A.7: Super Resolution with SRGAN
- Section A.8: Effect of Sparsity Granularity and Ratio

## A.1 Naïve Compression: StarGAN

The corresponding loss curves for the comparative experiment in Section 3 in the main paper are shown in Figure 1.

Figure 1: Loss curves of image-to-image translation pruning. **(a)**. Loss curve of StarGAN baseline. Loss curve of training the compressed generator from **(b)**. Self-Supervised fine-tuning **(ours)**, **(c)**. Smaller dense network, **(d)**. One-shot pruning and naive fine-tuning, **(e)**. Gradual pruning and naive fine-tuning, **(f)**. Gradual pruning during the initial training, **(g)**. One-shot pruning and distillation as fine-tuning, **(h)**. Gradual pruning and distillation as fine-tuning, **(i)**. AGP as fine-tuning and distillation without fine-tuning the original loss, **(j)**. Adversarial learning (fine-tuning), **(k)**. Knowledge distillation, **(l)**. Distillation on output of intermediate layers, **(m)**. E-M Quantization, and **(n)**. Prune both G and D models. **(a)**, **(c)** and **(f)** start from a randomly-initialized network at epoch 0, others pick up at the end of **(a)**.

Figures 2-4 show outputs of StarGAN compressed with various existing techniques (**c-n**), and the proposed self-supervised method (**b**). The baseline output is at the top (**a**) of each figure for comparison. Each row shows one input face translated to have black hair, blond hair, brown hair, the opposite gender, and a different age, and each row is a different method of compressing the network (the key is identical to that of Figure 1). Note that loss curves for (**h**) and (**m**) suggest good performance, but the actual results (Figure 1 and Table 1 in the main paper) are clearly worse, with FID scores of 38.985 and 159.767, respectively. (The baseline FID score is 6.113, and our method achieves 6.929.)

Figure 2: Example 1 of various approaches to compress StarGAN.

Figure 3: Example 2 of various approaches to compress StarGAN.

Figure 4: Example 3 of various approaches to compress StarGAN.

FID: 37.1296            34.2208            39.6708

Figure 5: Image synthesis on MNIST dataset with DCGAN pruned to 50% with fine-grained sparsity. Column 1: Handwritten numbers generated by the original generator, 2: Handwritten numbers generated by the generator pruned with our method, 3: Handwritten numbers generated by the pruned generator with traditional knowledge distillation adapted for GANs (Aguinaldo et al., 2019).

FID: 37.1296            48.3634            119.3063

Figure 6: Image synthesis on MNIST dataset with DCGAN of 75% fine-grained sparsity. Column 1: Handwritten numbers generated by the original generator, 2: Handwritten numbers generated by the generator pruned with our method, Column 3: Handwritten numbers generated by the pruned generator with traditional knowledge distillation adapted for GANs.

*FID: 37.1296*          *38.6406*          *42.6728*

Figure 7: Image synthesis on MNIST dataset with DCGAN pruned to 50% with filter-pruned sparsity. Column 1: Handwritten numbers generated by the original generator, 2: Handwritten numbers generated by the generator pruned with our method, 3: Handwritten numbers generated by the pruned generator with traditional knowledge distillation adapted for GANs (Aguinaldo et al., 2019).

*FID: 37.1296*          *118.3160*          *172.9123*

Figure 8: Image synthesis on MNIST dataset with DCGAN of 75% filter-pruned sparsity. Column 1: Handwritten numbers generated by the original generator, 2: Handwritten numbers generated by the generator pruned with our method, Column 3: Handwritten numbers generated by the pruned generator with traditional knowledge distillation adapted for GANs.

Figure 9: Image synthesis: from label maps to fake satellite images. Row 1: Original label maps, Row 2: Satellite images generated by the original generator, Row 3: Satellite images generated by the pruned generator, Row 4: Residual difference between generated images in Row 2 and 3 amplified by 10x.

Figure 10: Image synthesis: Two different random seeds, **unpruned**. Row 1: Original label maps, Rows 2-3: Generated fake satellite images by original generator trained with random seeds 15 and 63, Row 4: Residual difference between generated images in Row 2 and 3 amplified by 10x for higher contrast.

Figure 11: Image synthesis: from satellite images to fake label maps. Row 1: Original satellite images, Row 2: Label maps generated by the original generator, Row 3: Label maps generated by the pruned generator, Row 4: Residual difference between generated images in Row 2 and 3 amplified by 10x.

Figure 12: Image synthesis: Two different random seeds, **unpruned**. Row 1: Original satellite images, Rows 2-3: Generated fake label maps by original generator trained with random seeds 15 and 63, Row 4: Residual difference between generated images in Row 2 and 3 amplified by 10x.

Figure 13: Style transfer: from Monet to real photo style. Row 1: Original artwork images from Monet, Row 2: photographic style applied by the original generator, Row 3: photographic style applied by the compressed generator, Row 4: Residual difference between style transferred images in Row 2 and 3, amplified by 10x.

Figure 14: Style transfer: from real photo to Monet artwork style. Row 1: Original photos, Row 2: Monet's style applied by the original generator, Row 3: Monet's style applied by the compressed generator, Row 4: Residual difference between style transferred images in Row 2 and 3 amplified by 10x.

Figure 15: Image-to-image translation experiment: from real zebra images to fake horse images. Row 1: Original real zebra images, Row 2: Corresponding translated horse images by original generator, Row 3: Translated horse images by compressed generator, Row 4: Residual difference between translated images in Row 2 and 3 amplified by 10x.

Figure 16: Image-to-image translation experiment: from real horse images to fake zebra images. Row 1: Original real horse images, Row 2: Corresponding translated zebra images by original generator, Row 3: Translated zebra images by compressed generator, Row 4: Residual difference between translated images in Row 2 and 3 amplified by 10x.

Figure 17: Image-to-image translation example 1: facial attribute translation. Columns: 1. Original facial images, 2-4. Translated images to (black, blond, brown) hair colors, 5. Translated images to other gender, 6. Translated images to other age. Rows: Images translated by 1. original generator and 2. compressed generator, 3. Residual difference between Rows 1 and 2 amplified by 10x.

Figure 18: Image-to-image translation example 2: facial attribute translation. Columns: 1. Original facial images, 2-4. Translated images to (black, blond, brown) hair colors, 5. Translated images to other gender, 6. Translated images to other age. Rows: Images translated by 1. original generator and 2. compressed generator, 3. Residual difference between Rows 1 and 2 amplified by 10x.

Figure 19: Image-to-image translation example 3: facial attribute translation. Columns: 1. Original facial images, 2-4. Translated images to (black, blond, brown) hair colors, 5. Translated images to other gender, 6. Translated images to other age. Rows: Images translated by 1. original generator and 2. compressed generator, 3. Residual difference between Rows 1 and 2 amplified by 10x.

Figure 20: Image-to-image translation example 4: facial attribute translation. Columns: 1. Original facial images, 2-4. Translated images to (black, blond, brown) hair colors, 5. Translated images to other gender, 6. Translated images to other age. Rows: Images translated by 1. original generator and 2. compressed generator, 3. Residual difference between Rows 1 and 2 amplified by 10x.

Figure 21: Super resolution experiment. Column 1: Original high resolution images, Columns 2-4: Corresponding generated real high resolution images by original, filter-compressed, element-compressed generators. Each second row provides a detailed view of boxed regions.

**Pix2Pix**: map to satellite domain translation.

Figure 22: Domain translation: filter pruning to different sparsity levels. Row 1: Output of the baseline generator. Rows 2-5: Synthesized satellite images by generators pruned to sparsities of 25%, 50%, 75%, 90%.

Figure 23: Domain translation: fine-grained pruning to different sparsity levels. Row 1: Output of the baseline generator. Rows 2-5: Synthesized satellite images by generators pruned to sparsities of 25%, 50%, 75%, 90%.

**CycleGAN**: photographic style applied to Monet's paintings.

Figure 24: Style transfer: filter pruning to different sparsity levels. Row 1: Output of the baseline generator. Rows 2-5: Generated real photo style images by generators pruned to sparsities of 25%, 50%, 75%, 90%.

Figure 25: Style transfer: fine-grained pruning to different sparsities. Row 1: Output of the baseline generator. Rows 2-5: Generated real photo style images by generators pruned to sparsities of 25%, 50%, 75%, 90%.

**CycleGAN**: style of Monet's paintings applied to photographs.

Figure 26: Style transfer: filter pruning to different sparsity levels. Row 1: Output of the baseline generator. Rows 2-5: Generated real photo style images by generators pruned to sparsities of 25%, 50%, 75%, 90%.

Figure 27: Style transfer: fine-grained pruning to different sparsities. Row 1: Output of the baseline generator. Rows 2-5: Generated real photo style images by generators pruned to sparsities of 25%, 50%, 75%, 90%.

**CycleGAN**: zebra to horse image-to-image translation.

Figure 28: Image-to-image translation: filter pruning to different sparsities. Row 1: Baseline generator output. Rows 2-5: Generated real photo style images by generators pruned to sparsities of 25%, 50%, 75%, 90%.

Figure 29: Image-to-image translation: fine-grained pruning to different sparsities. Row 1: Baseline generator output. Rows 2-5: Generated real photo style images by generators pruned to sparsities of 25%, 50%, 75%, 90%.

**CycleGAN**: horse to zebra image-to-image translation.

Figure 30: Image-to-image translation: filter pruning to different sparsities. Row 1: Baseline generator output. Rows 2-5: Generated real photo style images by generators pruned to sparsities of 25%, 50%, 75%, 90%.

Figure 31: Image-to-image translation: fine-grained pruning to different sparsities. Row 1: Baseline generator output. Rows 2-5: Generated real photo style images by generators pruned to sparsities of 25%, 50%, 75%, 90%.

**StarGAN**: facial attribute image-to-image translation.

Figure 32: Image-to-image translation example 1: filter pruning to different sparsities. Row 1: Baseline generator output. Rows 2-5: Facial attribute translated images by generators pruned to sparsities of 25%, 50%, 75%, 90%.

Figure 33: Image-to-image translation example 1: fine-grained pruning to different sparsities. Row 1: Baseline generator output. Rows 2-5: Facial attribute translated images by generators pruned to sparsities of 25%, 50%, 75%, 90%.

Figure 34: Image-to-image translation example 2: filter pruning to different sparsities. Row 1: Baseline generator output. Rows 2-5: Facial attribute translated images by generators pruned to sparsities of 25%, 50%, 75%, 90%.

Figure 35: Image-to-image translation example 2: fine-grained pruning to different sparsities. Row 1: Baseline generator output. Rows 2-5: Facial attribute translated images by generators pruned to sparsities of 25%, 50%, 75%, 90%.

Figure 36: Super resolution: fine-grained pruning to different sparsities. Columns 1-4: Corresponding generated real high resolution images by generators pruned to sparsities of 25%, 50%, 75%, 90%.

The loss curves for the comparative experiment in Figure 32 and 34 are shown in Figure 37.

Figure 37: Loss curves of image-to-image translation experiments of filter pruning to different sparsities. **(a)**-**(d)**: Corresponding loss curve of the generator pruned to sparsities of 25%, 50%, 75%, 90%.

The loss curves for the comparative experiment in Figure 33 and 35 are shown in Figure 38.

Figure 38: Loss curves of image-to-image translation experiments of fine-grained pruning to different sparsities. **(a)**-**(d)**: Corresponding loss curve of the generator pruned to sparsities of 25%, 50%, 75%, 90%.