[Reviews · NeurIPS 2020]

Review 1

Summary and Contributions: The paper proposes an algorithm to train sparse generative networks. This is done by an adversarial learning framework, where a sparse network is trained jointly with a dense network, such that outputs of the sparse network are close to that of the dense network. Contributions: 1. The paper shows that existing pruning/sparsifying algorithms fail when applied to training generative models. 2. The proposed algorithm achieves 90% sparsification on certain image processing tasks. ---------------------------Edit after author feedback----------------------- I have read the author feedback and other reviews. I am raising my score to a 6 as my concern about quality of baselines was satisfactorily addressed in the author feedback, i.e., there do not exist other algorithms for compressing generative models, and hence the baselines considered are OK. However, my previous concerns about vagueness of discussion and strong unverified claims still remain. ------------------------------------------------------------------------------------

Strengths: + The algorithm has good empirical performance, as it can achieve 90% sparsification without loss in image quality. + The algorithm is easy to implement.

Weaknesses: - While the algorithm achieves good performance, the baselines are not convincing. The authors borrow sparsification algorithms designed for image classification networks, and report the performance of these algorithms when used for training GANs. As GANs are trained in an adversarial fashion, it seems natural that the proposed adversarial learning framework will produce better results than algorithms meant for image classification networks. - The algorithm is a simple modification of the StarGAN algorithm, where a sparse/compressed generative network is trained jointly with the dense network. - Some of the discussions about distributional entropies and the difficulties of using baseline algorithms do not have valid arguments to back them up. Right now they sound more like wild conjectures than intuition.

Correctness: The proposed claims and methods are correct, up to some minor issues outlined under "Weaknesses".

Clarity: - Section 4 needs more elaboration on all the loss functions. The notation is extremely confusing, and most of them are not defined. The ones that are defined are done verbally. The authors should not assume that a reader is familiar with what may be standard loss functions in the training of StarGAN, and should explicitly state what these functions are.

Relation to Prior Work: The relationship to prior work can be improved. Currently it seems that all the considered baselines are valid algorithms for generative modelling, although they are algorithms for training classifiers that have been used to train generative models. A better discussion on existing algorithms for generative modelling would be beneficial.

Reproducibility: Yes

Additional Feedback:


Review 2

Summary and Contributions: This paper deals with the problem of compressing the generator component of GANs. The authors outline why previous approaches to compression (either pruning or GAN-targeted approaches) don’t work, both qualitatively and quantitatively, and show a whole host of results of their compression technique on a variety of GAN tasks such as image synthesis, domain translation, and super resolution.

Strengths: 1. Paper is well-motivated and well-written and compressing generators for GANs is a practically useful problem 2. Idea is simple 3. As far as I know the proposed idea is novel, as it applies to compression of GANs 4. I like the speculation of reasons why existing compression techniques work for models trained on classification tasks but not GANs. While the authors do not explicitly test or verify any of the hypothesized reasons, it motivates the proposed approach and gives direction for future work

Weaknesses: 1. Quantitative experiments motivating the use of this method over existing compression methods are limited to StarGAN on CelebA. While there are a wealth of additional experiments for their method specifically, would be nice to provide the same level of thorough quantitative and qualitative results comparing to the baseline techniques for at least one more GAN architecture and dataset so that we can be confident that the method introduced isn’t overfit specifically to StarGAN and CelebA 2. The authors may be overloading the term “self-supervised”. The explanation for using this term is “Since the original discriminator is used as a proxy for a human’s subjective evaluation, we refer to this as ‘self-supervised’ compression” however (and perhaps there is a precedent for this that I am unaware of, but) self-supervision is usually a property of the dataset and not the models used. For example, image colorization (https://arxiv.org/pdf/1603.08511.pdf), inpainting (https://arxiv.org/pdf/1604.07379.pdf), and predicting image rotations (https://arxiv.org/pdf/1803.07728.pdf) all are examples of self-supervised learning tasks. Maybe there is a better term out there to describe this approach?

Correctness: The claims in the paper look to be correct and the paper doesn’t over-claim anything notable. The empirical methodology seems reliable because wherever possible the authors use publicly available implementations of different GAN training schemes that are open source on Github.

Clarity: This paper is very clear and readable. A few small notes to make it even more easy to read: - On line 78, for “dense baseline” would be nice to add “dense (i.e. uncompressed) baseline” for those who might not know what “dense baseline” means - Table 1 - in the caption put that these results are for StarGAN with the CelebA dataset. This is helpful for individuals who like to skim figures before deciding to read a paper, they can know what dataset the FID, etc. is measured on - Line 189: “there is no need to prune it to” → “there is no need to prune it too”

Relation to Prior Work: Prior work is discussed in Section 3 and differences between these and the proposed method are discussed.

Reproducibility: Yes

Additional Feedback: Overall I think this paper is well-written, the method described seems to work well, and the approach solves a problem that is practically useful (compression of generators in GANs). Aside from my reservations about using the term ‘self-supervised’ for this approach, I think this paper would make a nice addition to the NeurIPS paper lineup. ------------------- Post Rebuttal Feedback ------------------- In contrast to what some other reviewers thought, I don't think that it is unconvincing to use baselines from image classification if those are the only techniques that currently exist. I also don't necessarily agree that the paper requires more insightful understanding of why the method works in order to be considered a good paper - while more understanding would always of course be great to have, if a method is practically useful, then it should be used. Additionally, it is extremely common for GAN literature to be evaluated qualitatively with image samples. Because of this I will stick to my score.


Review 3

Summary and Contributions: This paper bases on some observations of the degradation in performance of existing work in compressing or distilling generative adversarial networks to propose some tricks to overcome the training issues.

Strengths: The experimental results look promising.

Weaknesses: For this type of papers, I expect more insightful diving into the problem rather than showing some generated images with some general comments. Without the insightful understanding, the proposed solution does not really convince me and seems to be some tricks rather than novel scientific discovery. The keyword “self-supervised” in the title also misleads me because I cannot see any pretext task that helps to learn and expose new features of the data targeting downstream tasks. The task what is doing in this paper is closer to model distillation for generative adversarial networks with some further tricks including: i) start from a well pre-trained discriminator rather than training from scratch and ii) new loss in Eqs. (5) and (7) to allow copying the full generator better.

Correctness: It seems to have no problem with theory and experiments.

Clarity: The writing style of the paper is hard to follow with abstract or common words when describing the drawbacks of previous approaches for example: - In Line 35: “In some cases, this result is masked by loss curves that look identical to the original training”. Why is it problematic? - In Line 166: “the new discriminator quickly falls into a low-entropy solution and cannot escape”. Again, why is it problematic? It means that the fake examples from a compressing generator are easily distinguishable to those of a full generator, does not it?

Relation to Prior Work: The prior work is clearly discussed in this paper.

Reproducibility: No

Additional Feedback: The idea is reasonable and applicable. However for this paper, I expect more insightful view and understanding of the training of existing approach for GAN compression. It would be better for me if you show clear evidence and explanation of what you claim: loss curves that look identical to the original training is not good and the new discriminator quickly falls into a low-entropy solution. Actually, in Section 3, the authors have done experiments on too many existing models without any specific focus on what they want to improve. This makes the discussion less condense and not deep enough. ----------------------- Post rebuttal: Thanks for your responses to my questions. However, I am still keen on my current score. The reasons include: i) Although the experiments are comprehensive and the paper also showed promising results, the discussion of the motivation of the paper is vague and less convincing to me without any remarkable supportive evidences. For example, the feedback: 'the discriminator falls into a low-entropy solution that will cause mode collapse' is very misleading and vague. It seems that you are talking about the low entropy of the distribution induced by the generator (the distribution over fake examples). ii) Furthermore, you argue that using a pre-trained discriminator and then doing fine-tuning is better than training a fresh new discriminator from scratch. However, what was provided only another misleading explanation: 'In some cases, this result is masked by loss curves that look identical to the original training as shown in (h) and (m)'. It is not clear to me why it is problematic. iii) Finally, in the experimental section, the experiments to compare two cases: i) training a new discriminator from scratch and ii) doing fine-tuning pre-trained discriminator should be conducted. However, there is no such kind of experiment and the performance of the proposed method was only showed. iv) Last but not least, optimization problems in (5), (6) have the form of A/B (B relates to the discriminator) which is usually hard to train. Probably the stop gradient was applied to consider B as a constant when training, but again this did not mention at all in the paper.


Review 4

Summary and Contributions: The authors proposed a new network compression method for GANs, specifically, pruning the generator in GAN. Through extensive experiments on different task and different networks, the authors demonstrated that the proposed method outperformed existing works both qualitatively and quantitatively. The method also achieved considerable speedup after pruning while maintaining generating data with good quality.

Strengths: 1. The authors utilized the power of the trained discriminator to guide the compression of the generator, which both outperformed existing methods and took a lot less time to train comparing to original training process. 2. The empirical evaluation of the proposed method is quite comprehensive. The authors performed extensive experiments to demonstrate the effectiveness of the propose method under different tasks with different networks. The methods were compared both qualitatively and quantitively. The authors also provided additional discussion on effects of different compression granularities and rates of the proposed method. 3. The authors clearly explained the motivation of the proposed method and provided detailed discussion of the shortcomings of existing approaches in generator compression in complex GAN tasks.

Weaknesses: 1. One main contribution of the paper is to utilize the trained discriminator to guide the network pruning process. While the authors demonstrated the effectiveness of this approach through different experiments, it would be helpful if the authors could provide more rigorous analysis or discussion on why using the trained discriminator in the original GAN could substantially boost the performance comparing to previous works. 2. In order to show the effectiveness of the compression approach, it would be helpful to compare training a small & dense network from scratch as (c) but with the discriminator initialized as the trained discriminator. The authors could include comparison in both the qualitative and quantitative results, as well as the training time with this setup. This could help further strengthen the argument of the effectiveness of the proposed method.

Correctness: The claims and methods in this paper are correct. The methodology is correct and intuitive.

Clarity: The paper is well written. Overall the methods are well explained, and the results are presented clearly. I appreciate the authors' discussion on the potential reasons of why existing approaches fails in complex GAN pruning tasks. However, it would be helpful if the author could elaborate more mathematically on the details of compression process in the proposed method during training.

Relation to Prior Work: The difference between the proposed method and existing works are clearly explained. The authors provided detailed discussion in section 3 about the reasons why existing methods produced unsatisfactory results when compressing the generator of a GAN in complex tasks. The main difference of the proposed method is the utilization of the trained discriminator.

Reproducibility: Yes

Additional Feedback:

[Author Response · NeurIPS 2020]

We thank the reviewers for their comments, which we found to be quite helpful. We have strengthened our submission with this feedback, and we believe the below addresses the main concerns identified in our submission.

**First** concern is the reviewer believed that we just borrow sparsification algorithms designed for image classification networks, and use them to train generative models. So he thought the baselines are not convincing.

We think this is a misunderstanding. For the experiments to compare with our method, we all follow the general settings and schedules when training the generative models, i.e., training in an adversarial fashion, as shown in **Table 1**. Just as comments from other reviewers, we apply the common sparsification algorithms in these baselines is to explain why existing techniques work for models trained on classification tasks but not GANs. So all the baselines are convincing.

**Second** concern is the reviewer thought we are missing the clear evidence of some claims about the loss curves.

Actually, the detailed evidence of loss curves are already provided in **Appendix**, **Section A.1**. And the detailed explains are already provided in **Discussion** part in **Section 3** of the manuscript. For example, in which situations, loss curves that look identical to the baseline training will also lead to the bad compression, and the discriminator falls into a low-entropy solution that will cause mode collapse.

**Third** concern is the reviewer thought we overload the term "self-supervised".

We thank the reviewers for pointing this out. We think the better term to describe our approach can be "self-tuning", "self-correcting", "autoregulative".

**Fourth** concern is the reviewer preferred more mathematically analysis on the performance boost in compression.

We provide the analysis from the **Bayes theory** perspective. The three deep neural networks in the **GAN** compression task are the original generative model $G_O$, the compressed generative model $G_C$, and the discriminative model $D$. Given $x$ as the input of the generative networks, we can denote the generative outputs as $G_O(x)$ and $G_C(x)$.

We use $x_i$ and $x_j$ to represent two training samples from different categories. Our target is to push closer the generative outputs of the original and compressed generative models with the samples from the same categories, while to push apart the outputs of these two models with the samples from different categories. **KL** divergence is applied to measure the difference between two generative representations. Ideally, the target can be denoted with the following formulas.

$$\textbf{KL}(G_O(x_i), G_C(x_i)) \to 0, \qquad \textbf{KL}(G_O(x_j), G_C(x_i)) \to \infty \tag{1}$$

We define a latent variable $S$ which represents whether the two input samples are from similar ($S = 1$) or different ($S = 0$) categories. For ease of notation, we define the event $U$ to denote the generative representations between the $G_O$ and $G_C$ models are similar, and the event $V$ denotes the $D$ model regards the generative results are similar, i.e.,

$$U \Rightarrow G_O(x) \doteq G_C(x), \qquad \overline{U} \Rightarrow G_O(x) \neq G_C(x)$$
$$V \Rightarrow D(G_O(x)) \doteq D(G_C(x)), \qquad \overline{V} \Rightarrow D(G_O(x)) \neq D(G_C(x)) \tag{2}$$

According to the total probability formula, for the whole GAN compression process:

$$P(S = 1) = P(S = 1 \,|\, U, V)P(U, V) + P(S = 1 \,|\, \overline{U}, V)P(\overline{U}, V)$$
$$+ P(S = 1 \,|\, U, \overline{V})P(U, \overline{V}) + P(S = 1 \,|\, \overline{U}, \overline{V})P(\overline{U}, \overline{V}) \tag{3}$$

If the discriminator is initialized by the well-trained model, then the probability of joint distribution for event $U$ and $V$ will be close to $P(U)$, while the probability of joint distribution for $\overline{U}, V$ and $U, \overline{V}$ will be close to 0, simplify **(3)** as:

$$P(S = 1) = P(S = 1 \,|\, U)P(U) + P(S = 1 \,|\, \overline{U})P(\overline{U}) \tag{4}$$

Because the $G_C$ model is initialized by $G_O$, so the second item in **formula (4)** has much less influence.

If the discriminator is randomly initialized as the original GAN baseline, then $U$ and $V$ can be regarded as the relative independent events. So the four items in **formula (3)** have a certain probability of occurrence.

Because the first item in **formula (3)** and **(4)** is our learning target. Our proposed method keeps the same total probability but changes the probability distribution. Because the optimization process during the learning cannot guarantee to find the global optimum. So an easier learning target has a higher expectation to be achieved during the same compression and optimization process. (We will extend to more rigorous prove without the one-page limitation.)

**Improvements:** We will add these minor improvements suggested by the reviewers in the final camera-ready version.

1. Provide the same level of thorough quantitative and qualitative results comparing to the baseline techniques for at least one more GAN architecture and dataset. (We will add them in **Appendix**.)

2. Improve the captions which will be helpful for readers who like to skim figures before deciding to read a paper.

3. Add the experiment setting as training a small & dense network from scratch, but with the discriminator initialized as the trained discriminator.

[Meta-Review · NeurIPS 2020]

The algorithm provides a method for compression of GANs, which has not been addressed previously. It is simple and has good empirical performance. While the paper provides strong empirical performance for a relevant task, the main weakness of the paper is that it lacks in terms of theoretical motivation and theoretical insights.